# Image quality of abdominal ultrasonography after esophagogastroduodenoscopy is preserved by using carbon dioxide insufflation: A non-inferiority test in the same subject

Tsuyoshi Suda[1¤a], Yukihiro Shirota[1]*, Hiroaki Takimoto[2], Yasunori Tsukada[3], Kensaku Takishita[3], Takahiro Nadamura[3], Masaki Miyazawa[1¤b], Yuji Hodo[1], Tokio Wakabayashi[1]

1 Department of Gastroenterology, Saiseikai Kanazawa Hospital, Kanazawa, Japan, 2 Medical Examination Center, Saiseikai Kanazawa Hospital, Kanazawa, Japan, 3 Department of Radiology, Saiseikai Kanazawa Hospital, Kanazawa, Japan

¤a Current address: Department of Gastroenterology, Japanese Red Cross Kanazawa Hospital, Kanazawa, Japan
¤b Current address: Department of Gastroenterology, Kanazawa University Hospital, Kanazawa, Japan
* shirota@r4.dion.ne.jp

## Abstract

Because bowel gas deteriorates the image quality of abdominal ultrasonography (AUS), it is common to perform AUS prior to esophagogastroduodenoscopy (EGD). This one-way order limits the availability of examination appointments. To evaluate whether EGD using insufflation of carbon dioxide ($CO_2$), which is rapidly absorbed by the gastrointestinal mucosa, preserves the image quality of AUS performed subsequently, we designed a non-inferiority test in which each subject underwent AUS, EGD with $CO_2$ insufflation, and a second AUS, in that order. All saved AUS moving images were randomized and imaging quality was evaluated at 16 organs using a four-point Likert-like scale that divides the depiction rate by 25%. Sample size was calculated to be 26 using the following: non-inferiority margin of −0.40 corresponding to depiction rate of −10%, difference of means of 0.40, common standard deviation of 1.25, power of 90%, and 1-sided α-level of 0.025. We enrolled 30 subjects. The mean and 95% confidence interval (CI) of the image quality score of all 16 organs at pre- and post-EGD AUS in the 30 subjects were 3.54 [3.48–3.60] and 3.46 [3.39–3.52], respectively. The difference in the means was 0.08 of the scores, corresponding to a 2% depiction rate. The effect size was 0.172. The image quality of post-EGD AUS was not inferior, as demonstrated by the 97.5% CI of the difference, which did not cross the non-inferiority margin of −0.40. In conclusion, the use of $CO_2$ for insufflation in EGD does not cause much deterioration in the image quality of AUS performed subsequently. Therefore, it is permissible to perform EGD prior to AUS, which is expected to improve the efficiency of examination setup.

**Data Availability Statement:** All relevant data are within the article and its Supporting information files.

**Funding:** The authors received no specific funding for this work.

**Competing interests:** The authors have declared that no competing interests exist.

## Introduction

As it is the recommended first-line imaging study for specific clinical situations [1], abdominal ultrasonography (AUS) is commonly used as a diagnostic imaging test for patients with abdominal pain, a screening test for various types of basal disease, and for medical check-ups. However, ultrasonography does not always provide sufficient diagnostic information due to inadequate visualization of intra-abdominal structures, especially if bowel gas is present.

In endoscopy, gas is introduced to achieve adequate distension of the GI lumen for safe advancement of endoscopes and for careful visualization of the mucosa. Therefore, when AUS and esophagogastroduodenoscopy (EGD) are planned at the same time, it is common to perform AUS prior to EGD. This one-way order limits the availability of examination appointments and reduces the efficiency of examination. Room air, which is widely used for GI luminal distension, has the advantages of universal availability and low cost. However, room air is poorly absorbed by the GI tract and causes post-procedure pain related to distension. In contrast, carbon dioxide ($CO_2$) is rapidly absorbed by the GI mucosa, driving increased interest in its use as an insufflation agent for endoscopic procedures [2].

We hypothesized that the use of $CO_2$ insufflation in EGD would not deteriorate the image quality of AUS performed subsequently. Two trials associated with this hypothesis have been reported. As only a limited number of organs, such as the pancreas alone [3] or the pancreas, extrahepatic bile duct, and inferior pole of the right kidney [4], were available for observation by AUS, and the statistical analysis was inadequate in these trials in which no statistical analysis was done [3] or a significant difference test was done instead of a non-inferiority test [4], the results of the data analyses are insufficient to test our hypothesis. In addition, they reported partially negative results showing that, in about 27% of patients, the image quality of AUS was worse [3]. However, they also reported results based on which they concluded that the image quality of AUS was improved in some patients [4]. To prove this hypothesis, we designed a non-inferiority test [5] that compared image quality of AUS performed following EGD with $CO_2$ insufflation (post-EGD AUS) with that of AUS prior to EGD (pre-EGD AUS) in the same subject.

## Materials and methods

This prospective and observational study was conducted in compliance with Declaration of Helsinki at a single institute and was approved by the Institutional Review Board of Saiseikai Kanazawa Hospital (No. H28-19). All experiments were carried out in accordance with the approved study plan and relevant guidelines. The subjects were examinees aged ≥20 years who visited our institute for medical check-ups including both AUS and EGD between January and March 2017. Those with an underlying previous history of operations such as gastrectomy, pancreatectomy, hepatectomy, and cholecystectomy were excluded to avoid the possible effect of intestinal peristalsis on the results of AUS. Those contraindicated for antispasmodics were also excluded. Written informed consent was obtained from all subjects. Each subject underwent AUS, EGD with $CO_2$ insufflation instead of air under planned moderate sedation, and a second AUS, in this order, within the same day. After EGD, subjects were transferred to a recovery bed and rested until their level of consciousness had recovered to the same level as before endoscopy.

### Ultrasonography

Ultrasonography was performed by three sonologists using a Hitachi Preirus with a 1–5 MHz curvilinear transducer (EUP-C715; Hitachi, Ltd., Tokyo, Japan). One of the three sonologists was board-certified by the Japan Society of Ultrasonics in Medicine, and the other two

sonologists had six or eight years of clinical experience and had performed a minimum of 500 clinical AUS examinations per year. The same technologist performed both pre- and post-EGD AUS for a given subject to reduce systematic error. All AUS moving images were saved. The 16 evaluation organs were the pancreas (head, body, and tail), liver (right, left, and caudate lobe), gallbladder, common biliary duct, spleen, kidney (right and left), aorta, celiac artery (CA), superior mesenteric artery (SMA), portal vein (PV), and splenic vein (SV), according to the AUS Cancer Screening Criteria of the Japanese Society of Gastrointestinal Cancer Screening [6].

## Review of ultrasonograms

A method for objectively estimating the quality of AUS images has not been established. Previous studies [3, 4, 7–9] have used three- to five-point Likert scales such as 1—not interpretable, 2—barely interpretable, 3—adequate for interpretation but of poor quality, 4—interpretable and of average quality, 5—interpretable and of superior quality. In this study, we developed a more objective scale by replacing the abstract evaluation with the depiction rate as an interval scale suitable for statistical analysis [10]. All saved AUS moving images, which were marked only with a study code, were randomized and then evaluated. To reduce systematic error, four expert reviewers board-certified by the Japanese Society of Gastroenterology rated image quality in all moving images using a four-point Likert-like scale as an interval scale: 1—depiction rate of 0%–24%, 2–25%–49%, 3–50%–74%, 4–75%–100%. The mean of the evaluation score by the four reviewers was used as the final score for each organ in each subject. The reviewers were blinded to the time points of the moving images, including whether the images they were reviewing were obtained before or after EGD. The reviewers were also blinded to the identity of the sonologist who performed the scan, to the study subject being imaged, and to the scores of the other reviewers.

## Endoscopic procedure

All EGD procedures were performed by the same endoscopist who was board-certified by the Japan Gastroenterological Endoscopy Society. EGD was performed using a GIF-H290 scope (Olympus Medical Systems, Tokyo, Japan) with a $CO_2$ Regulation Unit (UCR) as the $CO_2$ insufflation system (Olympus Medical Systems). Prior to endoscopy, 10 mg of butyl scopolamine bromide, an antispasmodic, was injected intramuscularly and diazepam was used for planned moderated sedation in all subjects.

## Statistical analysis

The primary endpoint of the study was the difference in AUS image quality before and after EGD with $CO_2$ insufflation using a non-inferiority test [5]. Secondary end points included BMI, abdominal circumference, gender difference, and age. Sample size calculation was based on a pre-defined margin of non-inferiority for image quality score set at –0.40, which is the same as a depiction rate of –10%. In the absence of published data, this margin was selected because we consider a larger difference as clinically relevant. The difference of means between pre- and post-EGD AUS was assumed to be 0.40 in the absence of previous data. The common SD was also assumed to be 1.25 in the absence of previous data. With a power of 90% and a 1-sided α-level of 0.025, we estimated that 26 subjects were needed to show non-inferiority in one-sample mean of post-EGD AUS. To compensate for unforeseeable problems, we aimed to enroll a total of 30 subjects. Sample size calculation was performed with the trial-size package of "R Project for Statistical Computing".

**Table 1. Examinees' characteristics.**

| | |
|---|---|
| Examinee | 30 |
| Male | 20 |
| Female | 10 |
| Age (years) | 47.4 ± 7.8 (28–62) |
| BMI (kg/m$^2$) | 23.0 ± 3.2 (17.8–32.8) |
| BMI > 30 | 1 |
| BMI < 18.5 | 0 |
| Abdominal circumference (cm) | 80.5 ± 9.1 (60–104) |
| EGD procedure time (min) | 4.7 ± 1.2 (3–9) |
| Duration of period after EGD to post-EGD AUS (min) | 73.8 ± 12.0 (53–110) |

Data are number or mean ± SD (Range).

SD, standard deviation; BMI, body mass index; EGD, esophagogastroduodenoscopy; AUS, abdominal ultrasonography.

## Results

Thirty subjects were enrolled in the study and their characteristics are shown in Table 1. The mean body mass index (BMI) was 23.0 kg/m$^2$ and was >30 kg/m$^2$ in only 1/30 subjects. The mean procedure time of EGD was 4.7 min and the mean duration of the period after EGD to post-EGD AUS was 73.8 min (range, 53–110 min).

All 30 examinees underwent pre-EDG AUS and post-EGD AUS, and a total of 60 AUS moving images were obtained. The means and 95% confidence interval (CI) for the total image quality score for all 16 organs for pre- and post-EGD AUS were 3.54 [3.48–3.60] and 3.46 [3.39–3.52], respectively (Fig 1). The difference in the means was 0.08 of the score, which corresponds to 2% of the depiction rate. Calculated based on these data, the effect size (Cohen's $d$) was 0.172, which is less than the value of 0.2, which is considered to be negligible.

Fig 2 shows the result of the non-inferiority test of the post-EGD AUS in all 16 organs depicted as the difference in image quality score between the post- and pre-EGD AUS. Because the two-sided 95% CI of the post-EGD AUS did not cross the 0-outcome difference, it was

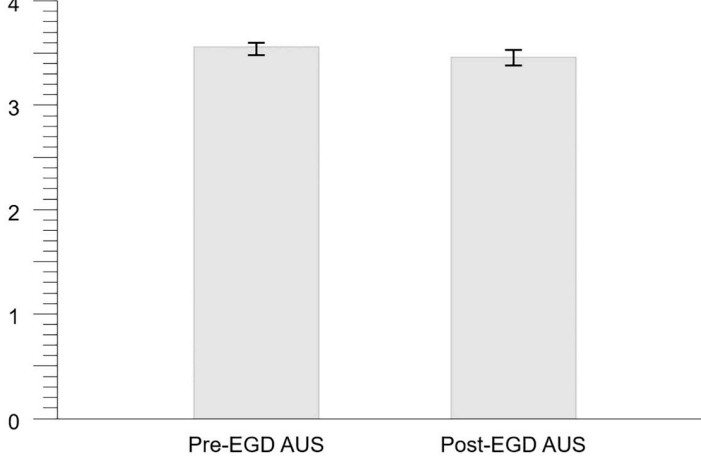

**Fig 1. Means and 95% CIs for total image quality score.** The means and 95% CI for the total image quality score for all 16 organs for pre- and post-EGD AUS are 3.54 [3.48–3.60] and 3.46 [3.39–3.52], respectively.

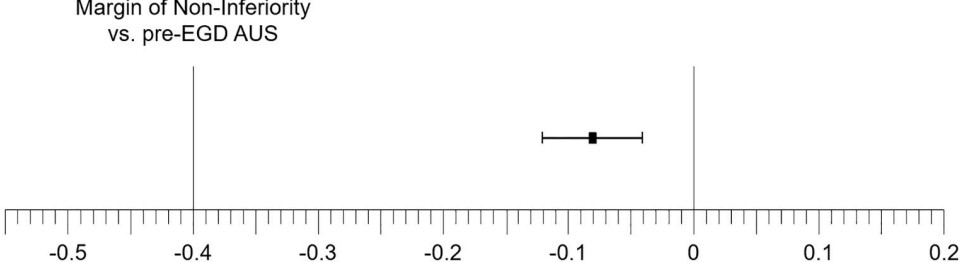

**Fig 2. Non-inferiority test of post-EGD AUS.** The result of the non-inferiority test of the post-EGD AUS in all 16 organs depicted as the difference in image quality score between the post- and pre-EGD AUS.

statistical significantly different to the pre-EGD AUS. However, because the entire CI of the post-EGD AUS was above the non-inferiority margin, it was non-inferior to the pre-EGD AUS.

Table 2 shows the means and 95% CIs of the image quality score for each organ at pre- and post-EGD AUS. Image quality scores for the pancreas tail, spleen, and celiac artery were <3, but those for other organs were >3. In all organs, the mean image quality for post-EGD AUS was less than that for pre-EGD AUS.

Fig 3 shows the results of the non-inferiority test of the post-EGD AUS in each organ, depicted as the difference in image quality score between the post- and pre-EGD AUS. In the celiac artery (CA) and superior mesenteric artery (SMA), non-inferiority of image quality of the post-EGD AUS could not be demonstrated when the margin was –0.40. In the other 14 organs, non-inferiority of image quality was demonstrated.

## Discussion

As well as its application as a diagnostic imaging test for patients with abdominal pain, AUS is also used for screening for metastasis in patients who have undergone resection for cancer

**Table 2. Means and 95% CIs of image quality score for each organ at pre- and post-EGD AUS.**

|  | pre-EGD AUS | post-EGD AUS |
|---|---|---|
| Pancreas Head | 3.38 [3.18–3.57] | 3.29 [3.01–3.57] |
| Pancreas Body | 3.63 [3.43–3.83] | 3.57 [3.35–3.78] |
| Pancreas Tail | 2.41 [2.14–2.67] | 2.35 [2.07–2.63] |
| Liver Right Lobe | 3.97 [3.93–4.01] | 3.89 [3.79–3.99] |
| Liver Left Lobe | 3.98 [3.96–4.01] | 3.94 [3.89–3.99] |
| Liver Caudate Lobe | 3.78 [3.64–3.91] | 3.64 [3.46–3.83] |
| GB | 3.97 [3.93–4.00] | 3.92 [3.84–3.99] |
| CBD | 3.37 [3.18–3.56] | 3.33 [3.07–3.58] |
| Spleen | 2.93 [2.81–3.06] | 2.93 [2.81–3.04] |
| Kidney Right | 3.93 [3.89–3.98] | 3.93 [3.88–3.97] |
| Kidney Left | 3.84 [3.78–3.90] | 3.82 [3.75–3.89] |
| Aorta | 3.73 [3.56–3.89] | 3.61 [3.38–3.84] |
| CA | 3.13 [2.75–3.52] | 2.98 [2.58–3.37] |
| SMA | 3.30 [2.98–3.62] | 3.05 [2.66–3.44] |
| PV | 3.81 [3.68–3.93] | 3.68 [3.50–3.87] |
| SV | 3.46 [3.26–3.65] | 3.38 [3.16–3.59] |

CI, confidence interval; GB, Gallbladder; CBD, common biliary duct; CA, celiac artery; SMA, superior mesenteric artery; PV, portal vein; SV, splenic vein.

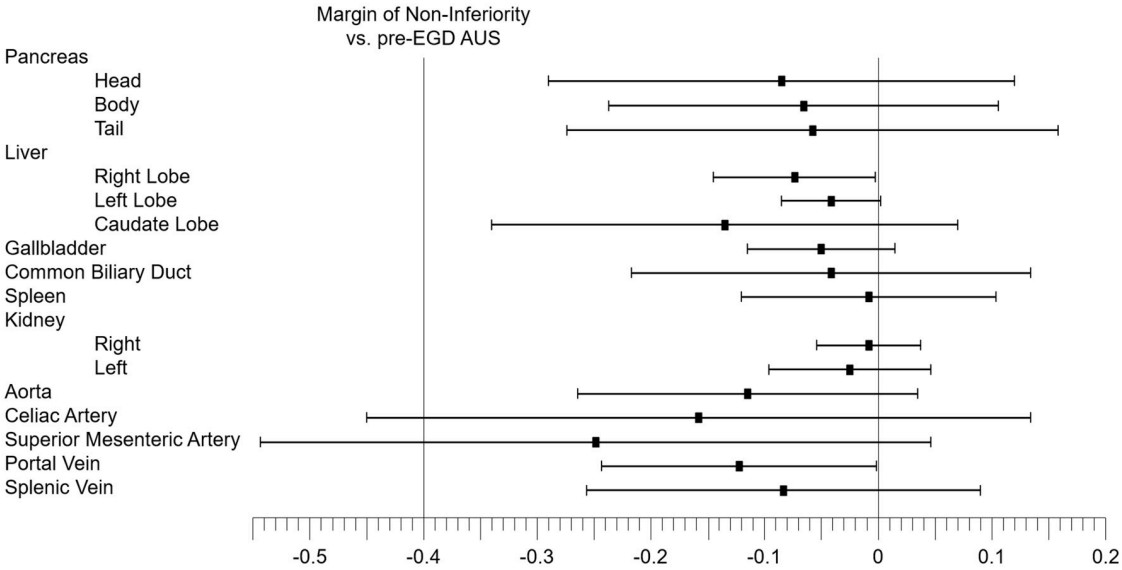

**Fig 3. Non-inferiority test of post-EGD AUS in each organ.** The results of the non-inferiority test of the post-EGD AUS in each organ, depicted as the difference in image quality score between the post- and pre-EGD AUS.

[11–13], hepatocellular carcinoma in patients with viral hepatitis [14], pancreatic cancer in patients at high risk due to family history, obesity, and type 2 diabetes [15], among others. AUS is simple to perform, involves no ionizing radiation, and is less costly than CT and MRI. EGD is used as a diagnostic tool for patients with symptoms of acid-peptic disease and also in screening for Helicobacter pylori-related gastric cancer [16, 17] and for follow-up after endoscopic resection of esophageal cancer [18] and gastric cancer [11, 19]. Both AUS and EGD are used in medical check-ups for asymptomatic patients with no basal disease. For patients who require both AUS and EGD, it is preferred that these procedures be scheduled on the same day because they both need preparation of nil per os. Because bowel gas deteriorates the image quality of AUS, it is common to perform AUS prior to EGD, and both must be completed within a limited period in the morning with efficient setup. This one-way order of AUS prior to EGD limits the availability of examination appointments and reduces the efficiency of examination. In Japan, the government introduced endoscopic screening for gastric cancer as a national program in 2016. It has been reported that the program was difficult to introduce immediately because of insufficient medical resources [20]. Increasing examination efficiency will become even more important in the future. In addition to endoscopy, there is concern that materials such as oral radiocontrast can deteriorate the image quality of AUS performed subsequently, and previous studies have investigated whether such materials affect the image quality of AUS from the perspective of a time-saving strategy in the emergent situation [7, 8].

Room air, which is widely used for GI luminal distension in EGD, has the advantages of universal availability and low cost. However, room air is poorly absorbed by the GI tract, resulting in post-procedure pain related to distension. As $CO_2$ is rapidly absorbed by the GI mucosa, there is increased interest in its use as an insufflation agent for endoscopic procedures. Many studies have shown the usefulness of $CO_2$ in endoscopic procedures that are relatively lengthy, including colonoscopy, EGD with colonoscopy, endoscopic retrograde cholangiopancreatography (ERCP), balloon endoscopy, and endoscopic procedures such as endoscopic submucosal dissection (ESD) [2].

We hypothesized that the use of $CO_2$ for insufflation in EGD would not deteriorate the image quality of subsequent AUS. To prove this hypothesis, we designed a non-inferiority test

and demonstrated a negligible difference in the means of image quality of all 16 organs between pre- and post-EGD AUS, as well as a non-inferiority of post-EGD AUS. However, in the non-inferiority test, the two-sided 95% CI of post-EGD AUS did not cross the 0-outcome difference, which indicates a statistically significant difference from the pre-EGD AUS. This puzzling phenomenon might have resulted from having a sample size that was too large, or/ and having too generous a non-inferiority margin [5]. An adequate sample size for this study had been calculated as around 26, but for a statistical study of all 16 organs in 30 cases, a significantly larger number of 480 samples was used. Regarding the non-inferiority margin, we set this at the point of –0.40, which is the same as a depiction rate of –10%, because we had considered a larger difference as clinically relevant. Furthermore, in the AUS studies after administration of oral radiocontrast, a one-point score difference in the five-point Likert scale was selected [7, 8]. Because the present –0.40-point margin of a four-point Likert-like scale is quite small compared with these reports, it is hard to say that our margin is too generous. Furthermore, because there is a possibility that the sample size was not adequate, we re-calculated the sample size using actual data obtained in the present study. Using a standard deviation (SD) of 0.69, which was calculated from the square root of the means of the sample variances of pre- and post-EGD AUS, the difference of means of 0.08, a power of 90%, and a 1-sided $\alpha$-level of 0.025, resulted in a sample size of 18 subjects. The number of 26 subjects that we had estimated before the study is not much different from this re-calculated sample size. Ultimately, the fact that the effect size was 0.172 also supports our hypothesis.

Regarding each organ, even if all of the means of image quality score for post-EGD AUS for each organ were slightly less than those for pre-EGD AUS, 14/16 organs were demonstrated to be non-inferior. The two organs that were not demonstrated to be non-inferior were the celiac artery and superior mesenteric artery. It is not clear why the AUS image quality deteriorated after EGD in these two arteries. Because they are located close to each other, observation of the site at which these arteries branch seems to be poor even if $CO_2$ insufflation is used. If the non-inferiority margin of –0.60, which is the same as a depiction rate of –15%, was acceptable rather than –0.40, all organs would be non-inferior.

We had expected that as $CO_2$ would extrude the bowel gas downstream and be absorbed rapidly, the AUS image quality after EGD with $CO_2$ insufflation would improve for some organs, such as the pancreas body and tail, where the quality is easily deteriorated by the presence of bowel gas in front of them. However, the mean image quality was reduced for all organs. Nakagawara et al. [4] reported that using $CO_2$ insufflation instead of air improved the image quality of AUS at 60 min after EGD in some patients. They used a three-point Likert scale of better, unchanged, and worse for evaluation of image quality of AUS after EGD, compared with that of AUS images before EGD as control. They reported that image quality for the pancreas head and body, pancreas tail, and extrahepatic bile duct were judged better in 26.1% (6/23 cases), 26.1% (6/23 cases), and 43.5% (10/23 cases), respectively, for AUS after EGD. When the method for evaluating the depiction rate in our study was changed to that of difference in image quality, of better/unchanged/worse, we found 11/6/13 (better by 36.7%), 8/ 9/13 (26.7%), 12/7/11 (40%), and 12/8/10 (40%) cases in the pancreas head, body, tail, and common bile duct, respectively. In the present study, the image quality also improved in some patients post-EGD AUS (Fig 4), but for other patients the quality worsened, so it is misleading to state that $CO_2$ insufflation improves AUS image quality even if this does occur in some patients. We concluded that $CO_2$ insufflation in EGD does not improve image quality in AUS that is performed following EGD.

Nakagawara et al. [4] also reported that image quality post-EGD AUS depended on the duration of the period after EGD. They showed that the deterioration in quality at <15 min after EGD had recovered by 30 min or later. In the present study, because EGD was performed

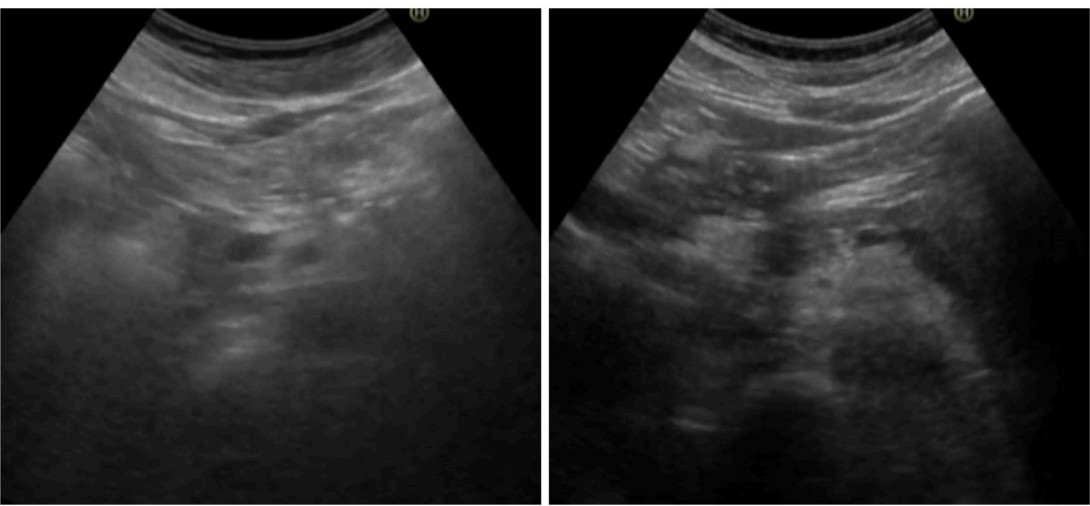

Pre-EGD AUS                                                              Post-EGD AUS

**Fig 4. Image quality of pancreas body improved in some patients post-EGD AUS.**

under planned moderate sedation, post-EGD AUS was performed 53–110 min after recovery from sedation. In the case of EGD performed without sedation, the shortest possible time between the end of EGD and AUS would be approximately 30 min.

Several limitations of the present study must be acknowledged. First, as the subjects were examinees for medical check-ups and enrolled in this study at our single institution, selection bias may have affected the results, particularly with regard to the subject characteristics. Second, in the absence of published data, the margin of non-inferiority was selected because we considered a larger difference to be clinically relevant. This margin size may have affected interpretation of the results. Third, because we demonstrated only the non-inferiority of the depiction rate, there is a possibility that EGD using $CO_2$ insufflation may affect the image characteristics of certain diseases in the subsequent AUS examination.

## Conclusions

The use of $CO_2$ for insufflation in EGD does not cause much deterioration in the image quality of AUS performed subsequently. Therefore, it is permissible to perform EGD prior to AUS, which is expected to improve the efficiency of examination setup.

## Supporting information

**S1 Table. Dataset of examinees' characteristics and Likert-like scales evaluated by the four reviewers for 16 organs in thirty subjects.**
(XLSX)

## Acknowledgments

We wish to thank Yoshimi Hirase for management of esophagogastroduodenoscopy practice using insufflation of carbon dioxide.

## Author Contributions

**Conceptualization:** Tsuyoshi Suda, Yukihiro Shirota, Hiroaki Takimoto.

**Data curation:** Tsuyoshi Suda, Yukihiro Shirota.

**Formal analysis:** Yukihiro Shirota.

**Investigation:** Tsuyoshi Suda, Yukihiro Shirota, Hiroaki Takimoto, Yasunori Tsukada, Kensaku Takishita, Takahiro Nadamura, Masaki Miyazawa, Yuji Hodo, Tokio Wakabayashi.

**Methodology:** Tsuyoshi Suda, Yukihiro Shirota, Hiroaki Takimoto.

**Project administration:** Tsuyoshi Suda.

**Resources:** Hiroaki Takimoto.

**Supervision:** Yukihiro Shirota.

**Visualization:** Yukihiro Shirota.

**Writing – original draft:** Yukihiro Shirota.

**Writing – review & editing:** Tsuyoshi Suda, Yukihiro Shirota, Hiroaki Takimoto, Yasunori Tsukada, Kensaku Takishita, Takahiro Nadamura, Masaki Miyazawa, Yuji Hodo, Tokio Wakabayashi.

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
