## [Decision Letter · Decision Letter 0]

12 Jul 2022

PONE-D-22-10775Image quality of abdominal ultrasonography after esophagogastroduodenoscopy is preserved by using carbon dioxide insufflation: A non-inferiority test in the same subjectPLOS ONE

Dear Dr. Shirota,

Thank you for submitting your manuscript to PLOS ONE. After careful consideration, we feel that it has merit but does not fully meet PLOS ONE’s publication criteria as it currently stands. Therefore, we invite you to submit a revised version of the manuscript that addresses the points raised during the review process.Please submit your revised manuscript by Aug 26 2022 11:59PM. If you will need more time than this to complete your revisions, please reply to this message or contact the journal office at plosone@plos.org. Please include the following items when submitting your revised manuscript:A rebuttal letter that responds to each point raised by the academic editor and reviewer(s). You should upload this letter as a separate file labeled 'Response to Reviewers'.A marked-up copy of your manuscript that highlights changes made to the original version. You should upload this as a separate file labeled 'Revised Manuscript with Track Changes'.An unmarked version of your revised paper without tracked changes. You should upload this as a separate file labeled 'Manuscript'.

We look forward to receiving your revised manuscript.

Kind regards,

Gopal Krishna Dhali, MBBS, MD, DM

Academic Editor

PLOS ONE

Journal Requirements:

Additional Editor Comments:

Dear authors,

I regret to inform you that the statistical reviewer has raised major concerns regarding your manuscript. Kindly find the detailed comments below and make necessary comments if you want us to reconsider the manuscript.

Reviewers' comments:

Reviewer's Responses to Questions

**Comments to the Author**

1. Is the manuscript technically sound, and do the data support the conclusions?

Reviewer #1: Yes

Reviewer #2: No

2. Has the statistical analysis been performed appropriately and rigorously? 

Reviewer #1: I Don't Know

Reviewer #2: No

3. Have the authors made all data underlying the findings in their manuscript fully available?

Reviewer #1: Yes

Reviewer #2: Yes

4. Is the manuscript presented in an intelligible fashion and written in standard English?

Reviewer #1: Yes

Reviewer #2: Yes

5. Review Comments to the Author

Reviewer #1: Yes, I feel that the authors have written the manuscript in an intelligible fashion and in standard English. However, there are a few typographical and grammatical errors which I have requested for corrections in a separate word file attachment. Regarding the statistical tools and setting up of the non-inferiority margin for the study, I do not consider myself to be competent enough and request the editor to take an opinion from a statistician.

Reviewer #2: PONE-D-22-10775: Statistical review

SUMMARY. This is a study to test whether the use of CO2 insufflation in esophagogastroduodenoscopy (EGD) does not deteriorate the image quality of abdominal ultrasonography (AUS) if this is performed subsequently. Although the research question is clearly stated, I have serious concerns about the statistical analysis which, in my opinion, should be fully revised: see major points 1-3 below. I also append a list of specific points that should be addressed.

MAJOR POINTS

1. Linkert scale. Although the outcome of interest is a continuous variable, the depiction rate, this value is reduced to a (ordinal) categorical variable through a Linkert scale. It doesn't seem a good idea, for at least three reasons. First, transforming a continuous variable to a categorical one is a waste of information and it is in general not recommended. Second, mean differences between Linkert-based scores are not mathematically sound (ordinal variables are better analyzed by quantiles) and, as a result, they are difficult to interpret (see also specific point 3 below). Third, Linkert-based scores are obviously not normally distributed and, as a result, conventional confidence intervals are inappropriate. If the authors are willing to use a Likert scale, this choice should be well motivated and the statistical analysis should be based on multinomial tests. Otherwise, I'd suggest to exploit the depiction rate as the outcome of interest: if it is normally distributed, it can be examined under traditional inference methods.

2. One side test. Although an analysis based on confidence intervals is not incorrect, a more natural approach to test non-inferiority would rely on one-side statistical tests where the null hypothesis is H_0: mean < a_0 and the alternative is H_1: mean > a_0 (a_0 being the desired threshold). Under this setting, p-values and powers (and sample sizes) are easier to compute and the results are easier to interpret.

3. Organ-specific differences. In a second step of the analysis, the sample is stratified according to the type of organ and the analysis is separately repeated withing each stratum (Table 2). Again, this approach is not totally incorrect, but it is based on an unnecessary waste of information due to the reduction of sample size. A more natural approach would rely on a regression (multinomial regression in the case of a Linkert-based categorical outcome or a more traditional regression in the case of a continuous dependent variable) where differences across organs is examined simultaneously.

4. Covariates. The available covariates are summarized in Table 1, but they are afterwards totally ignored in the subsequent analysis. Is it possible that image quality is somehow affected by these covariates? If it is, image quality differences should be adjusted by covariate value, under a traditional regression setting (see also major issue 3 for an another reason that would motivate a regression approach).

SPECIFIC POINTS

1. Figures 2 and 3 are totally black!

2. Depiction rate is the main outcome and should be precisely defined. How is it computed?

3. The Likert score is interpreted as the depiction rate divided by 4 (see lines 31 and 36). I don’t understand: the Linkert score clusters depictions rates into 4 intervals, hence the Linkert score is not necessarily the depiction rate divided by 4. Please clarify.

6. PLOS authors have the option to publish the peer review history of their article (what does this mean?). If published, this will include your full peer review and any attached files.

Reviewer #1: No

Reviewer #2: No

---

## [Author Response · Author response to Decision Letter 0]

30 Jul 2022

Point-by-point responses to the Editor and Reviewers

Editor

Comment: I regret to inform you that the statistical reviewer has raised major concerns regarding your manuscript. Kindly find the detailed comments below and make necessary comments if you want us to reconsider the manuscript.

Response: Thank you for your assessment. We have addressed the concerns of the reviewers, with special attention to the major concerns of the statistical reviewer (Reviewer #2), as noted below. 

Reviewer #1

Comment: I feel that the authors have written the manuscript in an intelligible fashion and in standard English. However, there are a few typographical and grammatical errors which I have requested for corrections in a separate word file attachment. Regarding the statistical tools and setting up of the non-inferiority margin for the study, I do not consider myself to be competent enough and request the editor to take an opinion from a statistician.

Response: Thank you for your assessment. We have addressed the points you raised.

Reviewer comments on the manuscript titled,” Image quality of abdominal ultrasonography after esophagogastroduodenoscopy is preserved by using carbon dioxide insufflation: A non-inferiority test in the same subject”.

Comment: It is always considered better if a patient can undergo USG abdomen (AUSG) and Esophagogastroduodenoscopy (EGD) on the same day one after another. This would reduce the problems associated with the delay in the allocation of dates for the two procedures on the same subject. However due to the air inside the stomach and the duodenum, following EGD, the USG abdomen cannot be done on account of poor visibility. It is hypothesized that this can be overcome if CO2 is used as the insufflating agent in place of air. In order to prove this hypothesis, the authors have pursued the study. The strengths of the study are 1) the adequate sample size of the study subjects and 2) the blinding of the interpreters of the USG images done by three experienced USG operators. I would however like to obtain a few responses from the authors regarding the following issues:

Response: Thank you for your assessment.

Comment 1: Figure 1 does not have any title, legend or label. The X-axis and the Y-axis should be labelled.

Response: Thank you. Your comment made us aware that we had made a mistake in creating the figure, and we have revised it accordingly.

Comment 2: The authors have mentioned the study type as a prospective one. However, there hasn’t been any follow-up of the subjects. Therefore it is a cross-sectional study and not a prospective one.

Response: A cross-sectional study is defined as a research design in which researchers collect data from many different individuals at a single point in time. In cross-sectional research, researchers observe variables without intervening. In a longitudinal study, researchers repeatedly examine the same individuals to detect any changes that might occur over a period of time. Longitudinal studies are a type of correlational research in which researchers observe and collect data on a number of variables without trying to affect the variables. Because we collected USG data at two different time points, and we intervened by performing EGD between them, our study is neither a cross-sectional nor a simple longitudinal study. Even though it was only a few hours, the subjects were followed up, so that our study can be considered a prospective study.

Comment 3: USG of an organ is an operator-dependent procedure. That some of the images following EGD appeared better to the interpreters as compared to the images done before EGD emphasizes this fact only. I would request the authors to comment on this fact with reference to the results of the study.

Response: As you point out, USG of an organ is an operator-dependent procedure. To reduce this systematic error, the same technologist performed both pre- and post-EGD USG in our study. In fact, we did not conclude that the image following EGD improved. Nakagawara et al. [4] reported that using CO2 insufflation instead of air improved the image quality of USG at 60 min after EGD in some patients. Contrary to their report, we concluded that CO2 insufflation in EGD does not improve image quality in USG that is performed following EGD.

Comment 4: P values with confidence intervals were not demonstrated while comparing the USG image quality scores of the different organs before and after EGD. I would request the authors to provide that. 

Response: We designed this as a non-inferiority study. As explained in reference 5 (Ahn S, Park SH, Lee KH. How to demonstrate similarity by using non-inferiority and equivalence statistical testing in radiology research. Radiology. 2013;267(2):328-38. doi: 10.1148/radiol.12120725.), neither a statistically significant difference between groups (P < .05) nor a lack of significant difference (P ≥ .05) from conventional statistical tests provides answers about equivalence/non-inferiority. Furthermore, because sample size was calculated for the non-inferiority test, not for a significant difference test, it is inappropriate to report p values in our study. 

Comment 5: There are a few typographical and grammatical errors which the authors need to correct. These are as mentioned below.

Line 89: Insert “and” between AUS and EGD.

Response: The sentence is “Each subject underwent AUS, EGD with CO2 insufflation instead of air under planned moderate sedation, and a second AUS, in this order, within the same day”. This can be summarized as “Each subject underwent AUS, EGD, and a second AUS, in this order”. What you suggested is that “Each subject underwent AUS and EGD, and a second AUS, in this order”, in which the order of the first AUS and EGD are not explicitly stated and are, therefore, slightly unclear. Therefore, we left this sentence as it is.

Line 97: “sonologists was” should be” sonologists were”.

Response: The sentence was “One of the three sonologists was board-certified by the Japan Society of Ultrasonics in Medicine”. In other words, “One sonologist was board-certified by the Japan Society of Ultrasonics in Medicine”. Therefore, we left this sentence as it is. 

Line 120: “was” should be “were”

Response: The sentence was “The mean of the evaluation score by the four reviewers was used as the final score for each organ in each subject”. This can be summarized as “The mean was used as the final score”. What you suggested is “The mean were used as the final score”, which is grammatically incorrect. Therefore, we left this sentence as it is.

Line 121: “as to” should be removed.

Response: We have revised the paper accordingly.

Changes: The following has been added to the revised manuscript on lines 120-122, and the revisions are shown in red.

The reviewers were blinded to the time points of the moving images, including as to whether the images they were reviewing were obtained before or after EGD. 

→

The reviewers were blinded to the time points of the moving images, including whether the images they were reviewing were obtained before or after EGD. 

Line 153: “mean” should be “the mean”

Response: We have revised the paper accordingly.

Changes: The following has been added to the revised manuscript on lines 152-153, and the revisions are shown in red.

Mean body mass index (BMI) was 23.0 kg/m2 and was >30 kg/m2 in only 1/30 subjects. Mean procedure time of EGD was 4.7 min and the mean duration of the period after EGD to post-EGD AUS was 73.8 min (range, 53–110 min).

→

The mean body mass index (BMI) was 23.0 kg/m2 and was >30 kg/m2 in only 1/30 subjects. The mean procedure time of EGD was 4.7 min and the mean duration of the period after EGD to post-EGD AUS was 73.8 min (range, 53–110 min).

Line 166: “considered to be” should be” is considered to be”.

Response: We have revised the paper accordingly.

Changes: The following has been added to the revised manuscript on lines 165-166, and the revisions are shown in red.

Calculated based on these data, the effect size (Cohen’s d) was 0.172, which is less than the value of 0.2 considered to be negligible.

→

Calculated based on these data, the effect size (Cohen’s d) was 0.172, which is less than the value of 0.2, which is considered to be negligible.

Line 214: “require” should be” requires.”

Response: For clarity, we have revised this sentence. 

Changes: The following has been added to the revised manuscript on lines 213-215, and the revisions are shown in red.

Patients who require both AUS and EGD prefer to schedule these procedures for the same day because they both need preparation of nil per os. 

→

For patients who require both AUS and EGD, it is preferred that these procedures be scheduled on the same day because they both need preparation of nil per os.

Line 247: for “a” statistical study.

Response: We have revised it accordingly.

Changes: The following has been added to the revised manuscript on line 247, and the revisions are shown in red.

An adequate sample size for this study had been calculated as around 26, but for statistical study of all 16 organs in 30 cases, a significantly larger number of 480 samples was used.

→

An adequate sample size for this study had been calculated as around 26, but for a statistical study of all 16 organs in 30 cases, a significantly larger number of 480 samples was used.

Line 248: samples “was” should be “were”

Response: The sentence is “An adequate sample size for this study had been calculated as around 26, but for statistical study of all 16 organs in 30 cases, a significantly larger number of 480 samples was used”. This can be summarized as, “An adequate sample size had been calculated as around 26, but for statistical study, a significantly larger number was used”. What you suggested is “a significantly larger number were used”, which is grammatically incorrect. Therefore, we left this sentence as it is.

Line 270: is same should be is “the” same.

Response: We have revised the paper accordingly.

Changes: The following has been added to the revised manuscript on line 270, and the revisions are shown in red.

If the non-inferiority margin of –0.60, which is same as a depiction rate of –15%, was acceptable rather than –0.40, all organs would be non-inferior.

→

If the non-inferiority margin of –0.60, which is the same as a depiction rate of –15%, was acceptable rather than –0.40, all organs would be non-inferior.

Line 287 image quality should be “the” image quality.

Response: We have revised the paper accordingly.

Changes: The following has been added to the revised manuscript on line 287, and the revisions are shown in red.

In the present study, image quality also improved in some patients post-EGD AUS (Fig 4), but for other patients the quality worsened, so it is misleading to state that CO2 insufflation improves AUS image quality even if this does occur in some patients.

→

In the present study, the image quality also improved in some patients post-EGD AUS (Fig 4), but for other patients, the quality worsened, so it is misleading to state that CO2 insufflation improves AUS image quality, even if it does occur in some patients.

Reviewer #2: PONE-D-22-10775: Statistical review

Comment: SUMMARY. This is a study to test whether the use of CO2 insufflation in esophagogastroduodenoscopy (EGD) does not deteriorate the image quality of abdominal ultrasonography (AUS) if this is performed subsequently. Although the research question is clearly stated, I have serious concerns about the statistical analysis which, in my opinion, should be fully revised: see major points 1-3 below. I also append a list of specific points that should be addressed.

Response: We recognize that there are some issues in the statistical analysis in which we differ. The concerns you have raised seem to be common, but we believe the two cited reports, references 5 and 10, have already resolved them. On the other hand, we do not believe it is possible to explain all statistical concepts in our paper, and we must assume that readers refer to these two references contained in our paper. Fortunately, PLOS ONE now offers accepted authors the opportunity to publish the peer review history, in which we would like to address readers who may have similar concerns here. We have addressed the points you have raised.

MAJOR POINTS

Comment:1. Linkert scale. Although the outcome of interest is a continuous variable, the depiction rate, this value is reduced to a (ordinal) categorical variable through a Linkert scale. It doesn't seem a good idea, for at least three reasons. First, transforming a continuous variable to a categorical one is a waste of information and it is in general not recommended. Second, mean differences between Linkert-based scores are not mathematically sound (ordinal variables are better analyzed by quantiles) and, as a result, they are difficult to interpret (see also specific point 3 below). Third, Linkert-based scores are obviously not normally distributed and, as a result, conventional confidence intervals are inappropriate. If the authors are willing to use a Likert scale, this choice should be well motivated and the statistical analysis should be based on multinomial tests. Otherwise, I'd suggest to exploit the depiction rate as the outcome of interest: if it is normally distributed, it can be examined under traditional inference methods.

Response: We agree that transforming a continuous variable to a categorical one is a waste of information. However, a method for evaluating the depiction rate of organs in AUS as continuous variable has not been established, and previous studies have used a Likert scale. To resolve this problem, we developed a more objective scale by replacing the abstract evaluation with the depiction rate as an interval scale more suitable for statistical analysis. We also considered that the minimum interval to distinguish the difference in the depiction rate subjectively by gastroenterologists who are familiar with AUS image evaluation is 25%. We mentioned this point in the Review of ultrasonograms in the Material and Methods section, as follows: A method for objectively estimating the quality of AUS images has not been established. Previous studies [3, 4, 7-9] have used three- to five-point Likert scales such as: 1 - not interpretable; 2 - barely interpretable; 3 - adequate for interpretation but of poor quality; 4 - interpretable and of average quality; and 5 - interpretable and of superior quality. In this study, we developed a more objective scale by replacing the abstract evaluation with the depiction rate as an interval scale suitable for statistical analysis [10]. All saved AUS moving images, which were marked only with a study code, were randomized and then evaluated. To reduce systematic error, four expert reviewers, board-certified by the Japanese Society of Gastroenterology, rated image quality in all moving images using a four-point Likert scale: 1 - depiction rate of 0%–24%; 2 - 25%–49%; 3 - 50%–74%; and 4 - 75%–100%. The mean of the evaluation score by the four reviewers was used as the final score for each organ in each subject. 

The second point, that mean differences between Likert-based scores are not mathematically sound, and the third point, that Likert-based scores are obviously not normally distributed and, as a result, conventional confidence intervals are inappropriate, are true, strictly speaking. However, what is left unsaid is how much the chance of an erroneous conclusion is increased. This is what statisticians call “robustness”, the extent to which the test will give the right answer even when assumptions are violated. This point is described in reference 10 (Norman G. Likert scales, levels of measurement and the "laws" of statistics. Adv Health Sci Educ Theory Pract. 2010;15(5):625-32. doi: 10.1007/s10459-010-9222-y.). Norman also said that “Parametric statistics can be used with Likert data, with unequal variances, and with non-normal distributions, with no fear of “coming to the wrong conclusion”. These findings are consistent with empirical literature dating back nearly 80 years. The controversy can cease (but likely won’t).” We designed our study based on this “robustness”.

Comment: 2. One side test. Although an analysis based on confidence intervals is not incorrect, a more natural approach to test non-inferiority would rely on one-side statistical tests where the null hypothesis is H_0: mean < a_0 and the alternative is H_1: mean > a_0 (a_0 being the desired threshold). Under this setting, p-values and powers (and sample sizes) are easier to compute and the results are easier to interpret. 

Response: For non-inferiority testing in which one is interested to know if the sensitivity (an index in which a larger value represents a better outcome) of the new test (PT) is not worse than the sensitivity of the active control test (PAC) without regard to its superiority to the active control test (H0: PT - PAC ≤ -Δ versus H1: PT - PAC > -Δ), only the relationship between the lower bound of the CI of PT - PAC and the non-inferiority margin (i.e., the lower bound of the equivalence range in this example) matters. Therefore, both one-sided and two-sided CIs of PT - PAC can be used for the analysis. This is described in reference 5 (Ahn S, Park SH, Lee KH. How to demonstrate similarity by using non-inferiority and equivalence statistical testing in radiology research. Radiology. 2013;267(2):328-38. doi: 10.1148/radiol.12120725.). They also state that neither a statistically significant difference between groups (P < .05) nor a lack of significant difference (P ≥ .05) from conventional statistical tests provides answers about equivalence/non-inferiority.

Comment: 3. Organ-specific differences. In a second step of the analysis, the sample is stratified according to the type of organ and the analysis is separately repeated withing each stratum (Table 2). Again, this approach is not totally incorrect, but it is based on an unnecessary waste of information due to the reduction of sample size. A more natural approach would rely on a regression (multinomial regression in the case of a Linkert-based categorical outcome or a more traditional regression in the case of a continuous dependent variable) where differences across organs is examined simultaneously.

Response: An adequate sample size for our non-inferiority test had been calculated as around 26. We enrolled 30 cases, and evaluated each organ in each case. This point was repeatedly described in our paper. Thus, the evaluation of organ-specific differences does not reduce the sample size, as you point out. On the contrary, in the statistical study for the primary endpoint, all 16 organs in 30 cases, a much larger number of 480 samples was used, which resulted in a puzzling phenomenon that both non-inferiority and a significant difference were demonstrated (Fig. 2). We mentioned this point in the Discussion section. Because gastroenterologists interested in our study realize that the depiction rate in daily practice may differ for each organ, it is important to report the mean value and confidence interval for each organ. On the other hand, this study is not aimed at statistically examining the differences across organs, and the primary endpoint is to prove non-inferiority of post-EGD AUS. Therefore, this study was designed as a non-inferiority study, not as a regression study to describe relationships between variables by fitting a line to the observed data.

Comment: 4. Covariates. The available covariates are summarized in Table 1, but they are afterwards totally ignored in the subsequent analysis. Is it possible that image quality is somehow affected by these covariates? If it is, image quality differences should be adjusted by covariate value, under a traditional regression setting (see also major issue 3 for an another reason that would motivate a regression approach).

Response: Our study design was a non-inferiority test in the same subjects. Therefore, the subjects’ characteristics summarized in Table 1 do not affect the results for non-inferiority of post-EGD AUS; for that reason, the variates are not covariates. 

SPECIFIC POINTS

Comment: 1. Figures 2 and 3 are totally black!

Response: Thank you. Your comment made us aware that we had made a mistake in creating the figures, and we have revised them accordingly.

Comment: 2. Depiction rate is the main outcome and should be precisely defined. How is it computed?

Response: The depiction rate was not calculated. We mentioned this point in the Review of ultrasonograms in the Material and Methods section, as follows: A method for objectively estimating the quality of AUS images has not been established. Previous studies [3, 4, 7-9] have used three- to five-point Likert scales such as 1 - not interpretable, 2 - barely interpretable, 3 - adequate for interpretation but of poor quality, 4 - interpretable and of average quality, 5 - interpretable and of superior quality. In this study, we developed a more objective scale by replacing the abstract evaluation with the depiction rate as an interval scale suitable for statistical analysis [10]. All saved AUS moving images, which were marked only with a study code, were randomized and then evaluated. To reduce systematic error, four expert reviewers board-certified by the Japanese Society of Gastroenterology rated image quality in all moving images using a four-point Likert scale: 1 - depiction rate of 0%–24%; 2 - 25%–49%; 3 - 50%–74%; and 4 - 75%–100%. The mean of the evaluation score by the four reviewers was used as the final score for each organ in each subject. The reviewers were blinded to the time points of the moving images, including as to whether the images they were reviewing were obtained before or after EGD. The reviewers were also blinded to the identity of the sonologist who performed the scan, to the study subject being imaged, and to the scores of the other reviewers.

Comment: 3. The Likert score is interpreted as the depiction rate divided by 4 (see lines 31 and 36). I don’t understand: the Linkert score clusters depictions rates into 4 intervals, hence the Linkert score is not necessarily the depiction rate divided by 4. Please clarify. 

Response: We used the Likert scale as an interval scale. We mentioned this point in the Review of ultrasonograms in the Material and Methods section as follows: A method for objectively estimating the quality of AUS images has not been established. Previous studies [3, 4, 7-9] have used three- to five-point Likert scales such as 1 - not interpretable, 2 - barely interpretable, 3 - adequate for interpretation but of poor quality, 4 - interpretable and of average quality, 5 - interpretable and of superior quality. In this study, we developed a more objective scale by replacing the abstract evaluation with the depiction rate as an interval scale suitable for statistical analysis [10]. All saved AUS moving images, which were marked only with a study code, were randomized and then evaluated. To reduce systematic error, four expert reviewers board-certified by the Japanese Society of Gastroenterology rated image quality in all moving images using a four-point Likert scale: 1 - depiction rate of 0%–24%; 2 - 25%–49%; 3 - 50%–74%; and 4 - 75%–100%. The mean of the evaluation score by the four reviewers was used as the final score for each organ in each subject. So, it is possible that the image quality score evaluated using our 4-point Likert scale corresponds to the depiction rate.

---

## [Decision Letter · Decision Letter 1]

15 Aug 2022

PONE-D-22-10775R1Image quality of abdominal ultrasonography after esophagogastroduodenoscopy is preserved by using carbon dioxide insufflation: A non-inferiority test in the same subjectPLOS ONE

Dear Dr. Shirota,

Thank you for submitting your manuscript to PLOS ONE. After careful consideration, we feel that it has merit but does not fully meet PLOS ONE’s publication criteria as it currently stands. Therefore, we invite you to submit a revised version of the manuscript that addresses the points raised during the review process.

We look forward to receiving your revised manuscript.

Kind regards,

Gopal Krishna Dhali, MBBS, MD, DM

Academic Editor

PLOS ONE

Reviewers' comments:

Reviewer's Responses to Questions

**Comments to the Author**

1. If the authors have adequately addressed your comments raised in a previous round of review and you feel that this manuscript is now acceptable for publication, you may indicate that here to bypass the “Comments to the Author” section, enter your conflict of interest statement in the “Confidential to Editor” section, and submit your "Accept" recommendation.

Reviewer #1: All comments have been addressed

Reviewer #2: (No Response)

2. Is the manuscript technically sound, and do the data support the conclusions?

Reviewer #1: Yes

Reviewer #2: No

3. Has the statistical analysis been performed appropriately and rigorously? 

Reviewer #1: I Don't Know

Reviewer #2: No

4. Have the authors made all data underlying the findings in their manuscript fully available?

Reviewer #1: Yes

Reviewer #2: Yes

5. Is the manuscript presented in an intelligible fashion and written in standard English?

Reviewer #1: Yes

Reviewer #2: Yes

6. Review Comments to the Author

Reviewer #1: The authors have responded to my questions regarding the manuscript titled, “Image quality of abdominal ultrasonography after esophagogastroduodenoscopy is preserved by using carbon dioxide insufflation: A non-inferiority test in the same subject” satisfactorily.

There was however one comment to which I felt the authors did not explain clearly. This was the response to comment no 3. The authors said they did not conclude that the USG images improved after EGD using Co2 insufflation. Indeed, they did not conclude in that manner but this is also true that they wrote about their observation that the image quality improved in some patients after EGD using Co2 insufflation (page 14, lines 287,288) which I feel has no scientific explanation rather than the inherently subjective nature (operator dependence) of the investigation under question. This factor was bound to affect the assessment of the USG images in the study which was assessed based on the four-point Likert scale. However, they have rightly concluded that it is misleading to state that CO2 insufflation improves AUS image quality even if this does occur in some patients.

I feel that in the absence of a validated pre-existing tool for assessment of the differences in the quality of the two USG images done on the same subject in a short time gap, this paper has merit to be published, provided the statistical part (assessed by reviewer 2) is sound enough.

Reviewer #2: PONE-D-22-10775: Statistical review

In my previous review, I raised 4 major issues and 3 specific points. All specific points have been addressed and the authors have provided reasonable answers to major issues 2-4 (thanks). However, I can't accept their answer to major issue 1.

From the answer to specific point no 2, I understand the depiction rate was not computed but it was summarized by the levels of a Linkert scale. Under this setting, two important points raised in my previous review are still unsolved, namely

1) mean differences between Linkert-based scores are not mathematically sound (ordinal variables are better analyzed by quantiles) and, as a result, they are difficult to interpret

2) Linkert-based scores are obviously not normally distributed and, as a result, conventional confidence intervals are inappropriate.

The authors agree with me on these two points, but they claim that their methods are robust with respect to the violations of these assumptions, citing reference 10. It is possible that some methods of the paper are robust, but robustness must be quantified. Unfortunately, I don’t see a simple way to justify robustness under point 1 above, because, as the authors say, we don’t have the continuous value of the depiction rate. Point 2 would instead be addressed by an appropriate simulation study, which is the standard approach to evaluate robustness; citing reference 10 is not enough because the degree of robustness varies from one study to another.

I still believe that trying to prove robustness is not the easiest approach in this study. Acknowledging the qualitative nature of the Linkert scale and using multinomial-based methods (as I suggested in my previous review) is much simpler. Furthermore, robustness is usually invoked when there are not standard methods to avoid assumptions violation. However, multinomial methods are standard and available in many statistical packages.

7. PLOS authors have the option to publish the peer review history of their article (what does this mean?). If published, this will include your full peer review and any attached files.

Reviewer #1: No

Reviewer #2: No

---

## [Author Response · Author response to Decision Letter 1]

2 Sep 2022

Point-by-point responses to the Editor and Reviewers

Reviewer #1

Comment: The authors have responded to my questions regarding the manuscript titled, “Image quality of abdominal ultrasonography after esophagogastroduodenoscopy is preserved by using carbon dioxide insufflation: A non-inferiority test in the same subject” satisfactorily.

There was however one comment to which I felt the authors did not explain clearly. This was the response to comment no 3. The authors said they did not conclude that the USG images improved after EGD using CO2 insufflation. Indeed, they did not conclude in that manner but this is also true that they wrote about their observation that the image quality improved in some patients after EGD using CO2 insufflation (page 14, lines 287,288) which I feel has no scientific explanation rather than the inherently subjective nature (operator dependence) of the investigation under question. This factor was bound to affect the assessment of the USG images in the study which was assessed based on the four-point Likert scale. However, they have rightly concluded that it is misleading to state that CO2 insufflation improves AUS image quality even if this does occur in some patients. I feel that in the absence of a validated pre-existing tool for assessment of the differences in the quality of the two USG images done on the same subject in a short time gap, this paper has merit to be published, provided the statistical part (assessed by reviewer 2) is sound enough.

Response: We believe that the sentence “In the present study, image quality also improved in some patients post-EGD AUS (Fig 4), but for other patients the quality worsened, so it is misleading to state that CO2 insufflation improves AUS image quality even if this does occur in some patients” is necessary to disprove the claims of previous papers (reference 4) and to negate the hypothesis that CO2 insufflation could improve image quality, which we had expected at the time of planning our study and is of interest to the readers. Thank you for your valuable comments.

Reviewer #2

Comment: In my previous review, I raised 4 major issues and 3 specific points. All specific points have been addressed and the authors have provided reasonable answers to major issues 2-4 (thanks). However, I can't accept their answer to major issue 1.

From the answer to specific point no 2, I understand the depiction rate was not computed but it was summarized by the levels of a Linkert scale. Under this setting, two important points raised in my previous review are still unsolved, namely

1) mean differences between Linkert-based scores are not mathematically sound (ordinal variables are better analyzed by quantiles) and, as a result, they are difficult to interpret

2) Linkert-based scores are obviously not normally distributed and, as a result, conventional confidence intervals are inappropriate.

The authors agree with me on these two points, but they claim that their methods are robust with respect to the violations of these assumptions, citing reference 10. It is possible that some methods of the paper are robust, but robustness must be quantified. Unfortunately, I don’t see a simple way to justify robustness under point 1 above, because, as the authors say, we don’t have the continuous value of the depiction rate. Point 2 would instead be addressed by an appropriate simulation study, which is the standard approach to evaluate robustness; citing reference 10 is not enough because the degree of robustness varies from one study to another.

I still believe that trying to prove robustness is not the easiest approach in this study. Acknowledging the qualitative nature of the Linkert scale and using multinomial-based methods (as I suggested in my previous review) is much simpler. Furthermore, robustness is usually invoked when there are not standard methods to avoid assumptions violation. However, multinomial methods are standard and available in many statistical packages.

Response: Our present study was designed and conducted based on statistical methods described in “Reviews and Commentary” of Radiology (reference 5), the most authoritative journal in diagnostic imaging. In clinical research, some flexibility is required to find a compromise between limitations in sample collection (in the evaluation of image quality, the limitation is that there is no method other than the Likert scale) and statistical correctness. In this review, the multinomial regression method that you are recommending is not mentioned, whereas a non-inferiority test using the mean difference is recommended to demonstrate non-inferiority of the outcome of one group to the outcome of another group, because non-inferiority cannot be proven correctly by multinomial regression methods. Furthermore, use of the mean difference between ordinal variables, such as Likert-based scores or grading scores is accepted. In fact, a study that is mentioned in this review used a non-inferiority test using the mean difference between grading scores. Please see the contents in the Abstract of that study below.

Hausleiter J, Martinoff S, Hadamitzky M, Martuscelli E, Pschierer I, Feuchtner GM, et al. Image quality and radiation exposure with a low tube voltage protocol for coronary CT angiography results of the PROTECTION II Trial. JACC Cardiovasc Imaging. 2010;3(11):1113-23.

Abstract

Objectives: The purpose of this study was to evaluate image quality and radiation dose using a 100 kVp tube voltage scan protocol compared with standard 120 kVp for coronary computed tomography angiography (CTA).

Background: Concerns have been raised about radiation exposure during coronary CTA. The use of a 100 kVp tube voltage scan protocol effectively lowers coronary CTA radiation dose compared with standard 120 kVp, but it is unknown whether image quality is maintained.

Methods: We enrolled 400 nonobese patients who underwent coronary CTA: 202 patients were randomly assigned to a 100 kVp protocol and 198 patients to a 120 kVp protocol. The primary end point was to demonstrate noninferiority in image quality with the 100 kVp protocol, which was assessed by a 4-point grading score (1 = nondiagnostic, 4 = excellent image quality). For the noninferiority analysis, a margin of -0.2 image quality score points for the difference between both scan protocols was pre-defined. Secondary end points included radiation dose and need for additional diagnostic tests during follow-up.

Results: The mean image quality scores in patients scanned with 100 kVp and 120 kVp were 3.30 ± 0.67 and 3.28 ± 0.68, respectively (p = 0.742); image quality of the 100 kVp protocol was not inferior, as demonstrated by the 97.5% confidence interval of the difference, which did not cross the pre-defined noninferiority margin of -0.2. The 100 kVp protocol was associated with a 31% relative reduction in radiation exposure (dose-length product: 868 ± 317 mGy × cm with 120 kVp vs. 599 ± 255 mGy × cm with 100 kVp; p < 0.0001). At 30-day follow-up, the need for additional diagnostic studies did not differ (13.4% vs. 19.2% for 100 kVp vs. 120 kVp, respectively; p = 0.114).

Conclusions: A coronary CTA protocol using 100 kVp tube voltage maintained image quality, but reduced radiation exposure by 31% as compared with the standard 120 kVp protocol. Thus, 100 kVp scan protocols should be considered for nonobese patients to keep radiation exposure as low as reasonably achievable. (Prospective Randomized Trial on Radiation Dose Estimates of Cardiac CT Angiography in Patients Scanned With a 100 kVp Protocol [PROTECTION II]; NCT00611780).

The sentence in your comment “as the authors say, we don’t have the continuous value of the depiction rate” appears to potentially be the result of a misunderstanding. We stated that “A method for evaluating the depiction rate of organs in AUS as continuous variables has not been established, and previous studies have used a Likert scale. To resolve this problem, we developed a more objective scale by replacing the abstract evaluation with the depiction rate as an interval scale more suitable for statistical analysis”. Again, the Likert scale we used is not an ordinal scale like the conventional Likert scale. It is an interval scale, even if it cannot be measured finely. Variables measured by the interval scale are continuous variables. Because it seems that the use of the word “Likert scale” is misleading, we have changed it to “Likert-like scale” in the revised manuscript.

There also appears to be a potential misunderstanding regarding reference 10. The paper (reference 10) is not just a single study. This paper shows that many studies, dating back to the 1930s, consistently show that parametric statistics are robust with respect to violations of many assumptions. Please see the contents in the Abstract below.

Reference 10. Norman G. Likert scales, levels of measurement and the "laws" of statistics. Adv Health Sci Educ Theory Pract. 2010;15(5):625-32. doi: 10.1007/s10459-010-9222-y.

Abstract

Reviewers of research reports frequently criticize the choice of statistical methods. While some of these criticisms are well-founded, frequently the use of various parametric methods such as analysis of variance, regression, correlation are faulted because: (a) the sample size is too small, (b) the data may not be normally distributed, or (c) The data are from Likert scales, which are ordinal, so parametric statistics cannot be used. In this paper, I dissect these arguments, and show that many studies, dating back to the 1930s consistently show that parametric statistics are robust with respect to violations of these assumptions. Hence, challenges like those above are unfounded, and parametric methods can be utilized without concern for ‘‘getting the wrong answer’’.

Although our present study is a clinical one with limited collection of samples with fine measurements, we believe that our analysis is based on recommended statistical methods by the authoritative journal, and that we have made every effort to be statistically accurate.

Changes 1: The following has been added to the revised manuscript on lines 113-119, and the revisions are shown in red.

In this study, we developed a more objective scale by replacing the abstract evaluation with the depiction rate as an interval scale suitable for statistical analysis [10]. All saved AUS moving images, which were marked only with a study code, were randomized and then evaluated. To reduce systematic error, four expert reviewers board-certified by the Japanese Society of Gastroenterology rated image quality in all moving images using a four-point Likert scale: 1 - depiction rate of 0%–24%, 2 - 25%–49%, 3 - 50%–74%, 4 - 75%–100%.

→

In this study, we developed a more objective scale by replacing the abstract evaluation with the depiction rate as an interval scale suitable for statistical analysis [10]. All saved AUS moving images, which were marked only with a study code, were randomized and then evaluated. To reduce systematic error, four expert reviewers board-certified by the Japanese Society of Gastroenterology rated image quality in all moving images using a four-point Likert-like scale as an interval scale: 1 - depiction rate of 0%–24%, 2 - 25%–49%, 3 - 50%–74%, 4 - 75%–100%.

Changes 2: The following has been added to the revised manuscript on lines 28-30, and the revisions are shown in red.

All saved AUS moving images were randomized and imaging quality was evaluated at 16 organs using a four-point Likert scale that divides the depiction rate by 25%.

→

All saved AUS moving images were randomized and imaging quality was evaluated at 16 organs using a four-point Likert-like scale that divides the depiction rate by 25%.

Changes 3: The following has been added to the revised manuscript on lines 252-254, and the revisions are shown in red.

Because the present –0.40-point margin of a four-point Likert scale is quite small compared with these reports, it is hard to say that our margin is too generous.

→

Because the present –0.40-point margin of a four-point Likert-like scale is quite small compared with these reports, it is hard to say that our margin is too generous.

Changes 4: The following has been added to the revised manuscript on lines 393-394, and the revisions are shown in red.

S1 Table. Dataset of examinees' characteristics and Likert scales evaluated by the four reviewers for 16 organs in thirty subjects. 

→

S1 Table. Dataset of examinees' characteristics and Likert-like scales evaluated by the four reviewers for 16 organs in thirty subjects.

---

## [Decision Letter · Decision Letter 2]

13 Sep 2022

Image quality of abdominal ultrasonography after esophagogastroduodenoscopy is preserved by using carbon dioxide insufflation: A non-inferiority test in the same subject

PONE-D-22-10775R2

Dear Dr. Shirota,

We’re pleased to inform you that your manuscript has been judged scientifically suitable for publication and will be formally accepted for publication once it meets all outstanding technical requirements.

Kind regards,

Gopal Krishna Dhali, MBBS, MD, DM

Academic Editor

PLOS ONE

Additional Editor Comments (optional):

Reviewers' comments:

Reviewer's Responses to Questions

**Comments to the Author**

1. If the authors have adequately addressed your comments raised in a previous round of review and you feel that this manuscript is now acceptable for publication, you may indicate that here to bypass the “Comments to the Author” section, enter your conflict of interest statement in the “Confidential to Editor” section, and submit your "Accept" recommendation.

Reviewer #2: All comments have been addressed

2. Is the manuscript technically sound, and do the data support the conclusions?

Reviewer #2: (No Response)

3. Has the statistical analysis been performed appropriately and rigorously? 

Reviewer #2: (No Response)

4. Have the authors made all data underlying the findings in their manuscript fully available?

Reviewer #2: (No Response)

5. Is the manuscript presented in an intelligible fashion and written in standard English?

Reviewer #2: (No Response)

6. Review Comments to the Author

Reviewer #2: The authors have clarified a critical misunderstanding. In this revision, the authors remark that the Likert scale used in the paper is not the conventional Likert scale, but it is an interval scale. Under this setting, differences can be computed although they are approximated.

7. PLOS authors have the option to publish the peer review history of their article (what does this mean?). If published, this will include your full peer review and any attached files.

Reviewer #2: No

---

## [Editor Report · Acceptance letter]

20 Sep 2022

PONE-D-22-10775R2 

Image quality of abdominal ultrasonography after esophagogastroduodenoscopy is preserved by using carbon dioxide insufflation: A non-inferiority test in the same subject 

Dear Dr. Shirota:

I'm pleased to inform you that your manuscript has been deemed suitable for publication in PLOS ONE. Congratulations! Your manuscript is now with our production department. 

Kind regards, 

on behalf of

Dr. Gopal Krishna Dhali 

Academic Editor

PLOS ONE